# Optimal, fast, and robust inference of reionization-era cosmology with the 21cmPIE-INN

Benedikt Schosser[1,2⋆], Caroline Heneka[1] and Tilman Plehn[1,3]

**1** Institut für Theoretische Physik, Universität Heidelberg, Germany
**2** Astronomisches Rechen-Institut,
Zentrum für Astronomie der Universität Heidelberg, Germany
**3** Interdisciplinary Center for Scientific Computing (IWR), Universität Heidelberg, Germany

⋆ schosser@thphys.uni-heidelberg.de

## Abstract

**Modern machine learning will allow for simulation-based inference from reionization-era 21cm observations at the Square Kilometre Array. Our framework combines a convolutional summary network and a conditional invertible network through a physics-inspired latent representation. It allows for an efficient and extremely fast determination of the posteriors of astrophysical and cosmological parameters, jointly with well-calibrated and on average unbiased summaries. The sensitivity to non-Gaussian information makes our method a promising alternative to the established power spectra.**

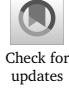

# 1    Introduction

Cosmic Dawn (CD) and the Epoch of Reionization (EoR) mark the emergence of the first galaxies and stars and the ionization of the intergalactic medium (IGM) by early luminous sources. Investigating these epochs, spanning redshifts from ∼5–6 to 20, helps us understand galaxy evolution, cosmological structure formation, the thermal history of the Universe, possible primordial sources of radiation, and the interplay between radiation, gas dynamics, and dark matter in the formation and evolution of observed structures.

An exciting way to explore the CD and EoR is through the redshifted 21cm line from the forbidden spin-flip transition of neutral hydrogen (HI). It is unique in its sensitivity to the spatial distribution of neutral hydrogen and the ionization state of the IGM and offers an exceptional avenue for mapping the large-scale structure. Experiments such as the Low Frequency Array (LOFAR) [1], the Murchison Widefield Array (MWA) [2], the Hydrogen Epoch of Reionization Array (HERA) [3], and the Precision Array for Probing the Epoch of Reionization (PAPER) [4] strive for a statistical detection of the 21cm signal, while the Square Kilometre Array (SKA)[1] promises 3D-tomography.

Intensity mapping of the 21cm line by SKA will allow investigations of dark energy and modifications of gravity [5–8], inflation [9,10], and dark matter [11–13]. Here, innovative analysis methods are essential, given the inherent non-Gaussianity, foreground contamination, and systematics. To simulate the 21cm signal during CD and EoR fast simulation frameworks are available for different astrophysical [14–17] and cosmological scenarios [18]. They are complemented by, albeit smaller, databases of radiative hydrodynamical simulations [19].

Recent progress in machine learning is transforming data-intensive analyses in fundamental physics and cosmology [20–22]. This is especially true when we can use simulations to relate fundamental parameters to observations and employ simulation-based inference [23–27]. Traditional simulation-based inference relies on pre-defined high-level observables, evaluated as one-dimensional or at most low-dimensional histograms. This bottleneck prevents us from using the full power of measurements or observations. The expected size of the SKA dataset, hundreds of Petabytes per year archived, and its complexity makes SKA a perfect example for the need to analyze data without this bottleneck.

For large cosmological surveys such as SKA we already know that convolutional neural networks (CNNs) outperform standard methods for source detection and characterization [28, 29], classification [30–32], and are able to jointly derive astrophysical and cosmological properties without summary statistic [33,34]. First steps towards simulation-based 21cm inference include variational inference [35] and direct density estimation [36–38], including alternative approaches like wavelet transforms [39,40]. We show how coupling the 3D-21cmPIE-Net feature extraction [41, 42] with a conditional invertible neural network (cINNs) [43, 44] allows for an optimal, fast, and robust inference of astrophysical and cosmological parameters. For the first time in this application, we jointly train these networks, enabling them to mutually refine their representations and achieve more reliable and well-calibrated posteriors.

We start by introducing the 21cm light cone dataset, the neural posterior estimation (NPE) method for simulation-based inference, the combination with the 3D-21cmPIE-Net feature extractor, the physics-inspired training protocol, and a sizeable range of validation and quality control methods in Sec. 2. In Sec. 3.1 we use marginalized 1-dimensional posteriors to control and confirm the calibration and the robustness of our 21cmPIE-INN setup. In Sec. 3.2 we show that its controlled, excellent performance remains when we add noise to the pure simulations for a realistic mock dataset. Finally, in Sec. 3.3 we show how the 21cmPIE-INN can extract a multi-dimensional posterior for astrophysics and cosmological parameters from a single light cone. Additional quality control measures and posteriors for more light cones are given in

---

[1] https://www.skatelescope.org/.

the Appendix.

## 2 Inference for 21cm tomography

Our goal is to use simulation-based inference to extract as much information as possible from complex 21cm data and avoid the bottlenecks of classic analysis methods. This requires an appropriate representation of the data, introduced in Sec. 2.1, the conditional generative neural network described in Sec. 2.2, a physics-inspired data pre-processing introduced in Sec. 2.3, a dedicated training protocol discussed in Sec. 2.4, and a detailed validation, Sec. 2.5. Our framework will then allow for a fast, amortized inference of cosmological and astrophysical parameters from a single 21cm light cone.

### 2.1 21cm light cone data

Our data consists of 5000 3D-light cones (LC) of 21cm brightness temperature fluctuations $\delta T_b(x, \nu)$, with on-sky coordinates $x$ and frequency $\nu$. The LCs are produced with the semi-numerical code 21cmFASTv3 [45].[2] It generates initial density and velocity conditions and evolves them at first and second order perturbation theory using the Zel'dovich approximation [46]. A region is flagged as ionized, if the fraction of collapsed matter, $f_{\mathrm{coll}}$, exceeds the inverse ionizing efficiency of star formation, $\zeta^{-1}$. The fraction $f_{\mathrm{coll}}$ is calculated in an excursion-set approach, where the density field is filtered with a top-hat of decreasing size. The code accounts for partially ionized regions with an ionized fraction $f_{\mathrm{coll}}\zeta$.

Besides the ionization fraction, the 21cm signal at higher redshifts crucially depends on the spin gas temperature $T_{\mathrm{S}}$, which in turn depends on couplings to kinetic gas temperature and density. We do not assume the so-called post-heating regime and instead fully evolve spin temperature boxes. To generate LCs, coeval cubes of 21cm brightness temperature fluctuations, evolved with redshift, are stitched together in the last step.

The resulting 21cm brightness fluctuations depend on several cosmological and astrophysical parameters. For our simple, proof-of-concept study we combine two parameters defining our cosmological model, two parameters describing astrophysics during cosmic dawn, and two parameters to account for EoR astrophysics [41]:

- Matter density $\Omega_{\mathrm{m}} \in [0.2, 0.4]$
  It controls structure formation, where a wide range encompasses the Planck limits [47].

- Warm dark matter mass $m_{\mathrm{WDM}} \in [0.3, 10]\,\mathrm{keV}$
  The conservative limit allows for a wide range of possible behavior, where the lower limit exhibits a tension with Cold Dark Matter (CDM), and current astrophysical constraints point towards values larger than a few keV [48, 49]. Here, structure formation looks more and more similar to CDM, as the free-streaming length is inversely proportional to the WDM mass.

- Minimum virial temperature $T_{\mathrm{vir}} \in [10^4, 10^{5.3}]\,\mathrm{K}$
  The minimum virial temperature needed for cooling within halos to enable star formation.

- Ionization efficiency $\zeta \in [10, 250]$
  It is represented by the composite parameter

$$\zeta = 30\frac{f_{\mathrm{esc}}}{0.3}\,\frac{f_\star}{0.05}\,\frac{N_{\gamma/b}}{4000}\,\frac{2}{1 + n_{\mathrm{rec}}}, \tag{1}$$

---

[2]https://github.com/21cmFAST/21cmFAST.

where factors such as the escape fraction of ionizing photons into the intergalactic medium $f_{\text{esc}}$, the fraction of galactic gas in stars $f_\star$, the number of ionizing photons per baryon in stars $N_{\gamma/b}$, and the typical number density of recombinations for hydrogen in the intergalactic medium $n_{\text{rec}}$ contribute to a versatile range of recombination scenarios;

- Specific X-ray luminosity $L_{\text{X}} \in [10^{38}, 10^{42}]\,\text{erg}\,\text{s}^{-1}\,\text{M}_\odot^{-1}\,\text{yr}$
  Integrated luminosity $< 2\,\text{keV}$ per unit star formation rate that escapes host galaxies.

- X-ray energy threshold for self-absorption by host galaxies $E_0 \in [100, 1500]\,\text{eV}$
  X-rays with energies below $E_0$ do not escape the host galaxy.

All other cosmological parameters are fixed to the Planck measurements [47], assuming flatness and a cosmological constant. This means $\Omega_{\text{b}} = 0.04897$, $\sigma_8 = 0.8102$, $h = 0.6766$, and $n_s = 0.9665$.

To generate our training data, we randomly sample parameters from flat priors and then simulate the corresponding light cone of 21cm brightness offset temperature. The box size is 200 Mpc at a resolution of 1.42 Mpc, and the redshift range simulated is $z = 5 \dots 35$. Each light cone has the shape of $(2350, 140, 140)$ voxels, keeping in mind that $\Omega_{\text{m}}$ impacts the length of each light cone in terms of redshift. Hence, only at $\Omega_{\text{m}} = 0.4$ the simulated light cone includes $z = 35$, light cones for smaller $\Omega_{\text{m}}$ stop at slightly lower redshifts, to keep the number of pixels in temporal or redshift direction fixed.

Due to the wide parameter ranges some of the simulated reionization histories and light cones are excluded by observations. We filter the light cones to exclude extreme reionization histories, requiring that the optical depth $\tau_{\text{reio}}$ is within $5\sigma$ of the Planck measurement $0.054 \pm 0.007$ [47], and that the IGM mean neutral fraction at redshift 5 is below 0.1. From 5000 valid light cones in our dataset, we use 3600 for training, 400 for validation, and 1000 for testing the network.

Going beyond the idealized, pure simulations, we generate mock observed light cones using 21cmSense [50, 51].[3] Our 5000 simulated light cones are transformed by coevally evolved simulation boxes at fixed redshifts. These boxes are split at certain redshift values, and thermal noise is calculated for each box using 21cmSense. The resulting noise is added to the Fourier-transformed box, and the mock light cone is reconstructed in real space. The thermal noise assumptions are based on 1080 hours of integrated SKA-Low stage 1 observations with specific instrument characteristics and baseline distribution. Three foreground settings (optimistic, moderate, pessimistic) in 21cmSense account for different scenarios, where the optimistic scenario considers the 21cm foreground wedge in $k$-space covering only the primary field-of-view of the instrument. In this work, we use this optimistic foreground scenario. Mock light cones with this noise setting are mainly affected at higher redshifts.

## 2.2 Neural posterior estimation

Conditional generative networks have shown great promise for Bayesian inference in fundamental physics and cosmology [52–55]. NPE allows for simulation-based inference without assuming a likelihood shape. It combines a summary network, to reduce simulated or observed data to an appropriate latent representation, with a conditional generative network. Specifically, we implemented the invertible neural network (INN) [43] version of a normalizing flow [56, 57], following the BayesFlow [58, 59] concept. Our implementation is available to use on GitHub. Conditional on an observation, this cINN generates the posterior in parameter space. For 21cm tomography, its input is the full signal light cones of the 21cm offset

---

[3]https://github.com/rasg-affiliates/21cmSense.

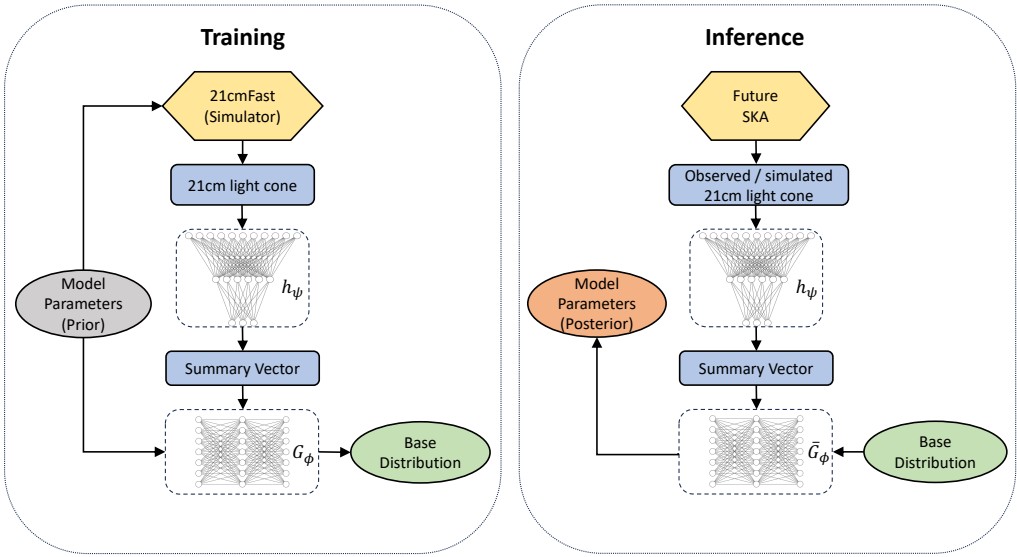

Figure 1: Illustration of Bayesian inference with conditional generative networks, following the BayesFlow concept.

brightness temperature, without reducing the 3D-maps to summary statistics, like the power spectrum.

This inference method is illustrated in Fig. 1. The two networks are trained on simulations, providing the model parameters and paired simulated data. The simulated data is passed through the summary network, to provide the condition for the cINN. This cINN maps the model parameters to a Gaussian latent distribution by minimizing a likelihood loss. For the inference, the summarized data is again the condition for the cINN, which now samples from the Gaussian to generate the posterior in model space.

The simulation uses the model parameters $\theta_{1:D}$ to generate data $x_{1:N}$, where the elements $x_i$ can be scalars or vectors. To simplify our notation, we first omit the summary network, so the data $x$ is fed directly to the cINN. This cINN links a latent Gaussian distribution and the posterior over model space [21],

$$\text{latent } r \sim \mathcal{N}_{0,1} \quad \underset{\leftarrow \overline{G}_\phi(\theta|x)}{\overset{G_\phi(r|x)\rightarrow}{\longleftrightarrow}} \quad \text{model space } \theta \sim p(\theta|x). \tag{2}$$

Here $\overline{G}_\phi(\theta|x)$ denotes the inverse transformation to $G_\phi(r|x)$, both encoded in the network parameters $\phi$. The training goal is to approximate the true posterior,

$$p_\phi(\theta|x) \approx p(\theta|x), \tag{3}$$

for all possible parameters $\theta$ and data $x$. To this end, we minimize the Kullback-Leibler (KL) divergence between the approximate and true posteriors

$$D_{\text{KL}}[p(\theta|x), p_\phi(\theta|x)] = \left\langle \log p(\theta|x) - \log p_\phi(\theta|x) \right\rangle_{p(\theta,x)}, \tag{4}$$

or the weight-dependent loss function

$$\mathcal{L}_{\text{cINN}} = -\left\langle \log p_\phi(\theta|x) \right\rangle_{p(\theta,x)}. \tag{5}$$

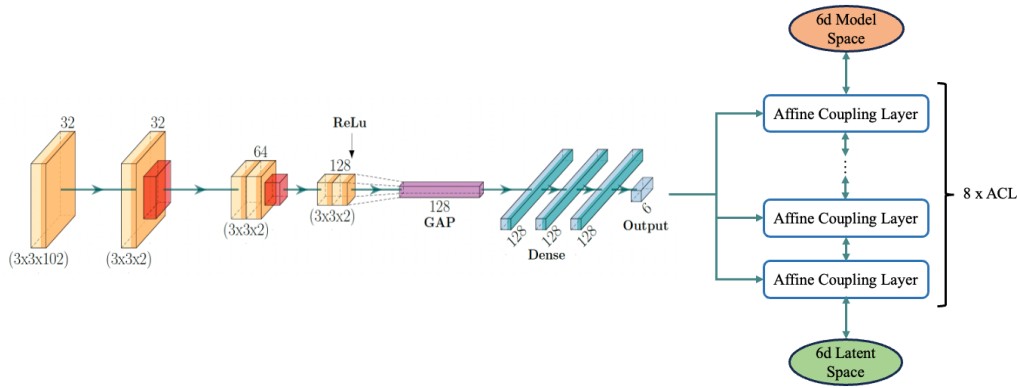

Figure 2: Schematic representation of the 21cmPIE-INN, combining the physics-inspired summary network [41] and the cINN.

This loss function is evaluated over pairs of model parameters and the corresponding simulated data. The posterior is encoded in the cINN through a Jacobian of $\overline{G}_\phi$,

$$p_\phi(\theta|x) = \mathcal{N}_{0,1}(\overline{G}_\phi(\theta|x)) \left| \frac{\partial \overline{G}_\phi(\theta|x)}{\partial \theta} \right|$$

$$\Rightarrow \quad \mathcal{L}_{\text{cINN}} = -\left\langle \log \mathcal{N}_{0,1}(\overline{G}_\phi(\theta|x)) + \log \left| \frac{\partial \overline{G}_\phi(\theta|x)}{\partial \theta} \right| \right\rangle_{p(\theta,x)}$$

$$= -\left\langle \frac{|\overline{G}_\phi(\theta|x)|^2}{2} + \log \left| \frac{\partial \overline{G}_\phi(\theta|x)}{\partial \theta} \right| \right\rangle_{p(\theta,x)}. \tag{6}$$

The first term regularizes the network, while the second term trains the Jacobian of the cINN.

The summary network $h_\psi$ transforms the input data before it enters the cINN as the condition. It does not have to be big and can be trained together with the cINN, using the NPE loss function

$$\mathcal{L}_{\text{NPE}} = -\left\langle \frac{|\overline{G}_\phi(\theta|h_\psi(x))|^2}{2} + \log \left| \frac{\partial \overline{G}_\phi(\theta|h_\psi(x))}{\partial \theta} \right| \right\rangle_{p(\theta,x)}. \tag{7}$$

The number of instances we train the network on is free, as long as we evaluate the network on the same number of instances of the observed or test data $x^o_{1:N}$.

The loss in Eq.(6) assumes that the Jacobian relating the model parameters $\theta$ and latent random variables $r$ can be evaluated fast [60]. The classic choice is a stack of affine coupling blocks [61] and rotational layers. These affine layers are simple and extremely fast. In case we need a more expressive invertible network, we can replace them by cubic [62] or rational quadratic splines [63], if needed with learned rotations [64] or periodic boundary conditions [65].

## 2.3 21cmPIE-INN

As described above, our summary network $h_\psi(x)$ compresses a large three-dimensional data object, the light cone. We choose a strong compression to the six parameters used for light cone simulation in CD and EoR astrophysics and cosmology, as described in Sec. 2.1. In this case, the network needs to be very expressive. We use a 3D-convolutional network (CNN), the 3D-21cmPIE-Net introduced for parameter regression [41, 42]. The 3D-21cmPIE-Net provides

Table 1: 3D-CNN and cINN architectures and hyperparameters.

|  | Layer | Shape |
|---|---|---|
| | Input Layer | (1,140,140,2350) |
| | 3x3x102 Conv3D | (32,138,138,23) |
| | 3x3x2 Conv3D | (32,136,136,22) |
| | 2x2x1 Max Pooling | (32,68,68,22) |
| | 3x3x2 Conv3D | (64,66,66,21) |
| | 1x1x0 Zero Padding | (64,66,66,20) |
| 3D-CNN | 3x3x2 Conv3D | (64,66,66,20) |
| | 2x2x1 Max Pooling | (64,33,33,20) |
| | 3x3x2 Conv3D | (128,31,31,19) |
| | 1x1x0 Zero Padding | (128,33,33,19) |
| | 3x3x2 Conv3D | (128,31,31,18) |
| | Global Average Pooling | (128) |
| | 3 x Dense | (128) |
| | Dense | (6) |
| | Number of inferred parameters | 6 |
| cINN | Coupling layers | 8 |
| | Fully connected coupling layer architecture | 256, for all layers |

fast and optimal convergence at a moderate size of the required training dataset, compared to alternative networks [36,37]. It has been shown to efficiently provide unbiased parameter estimates for both astrophysical and cosmological parameters, outperforming for example larger Long Short Term Memory networks, and requiring smaller training datasets. The architecture is summarized in Tab. 1 and schematically shown in Fig. 2. The asymmetric $(3 \times 3 \times 102)$-kernel of the first filter reflects the difference between fluctuations in temporal $z$-direction and spatial direction. The $(1 \times 1 \times 102)$-stride reduces the dimensionality of the following layers, while still capturing the relevant physics. The kernel size of the hidden layers is set to $(3 \times 3 \times 2)$ and max pooling layers are applied only in the spatial direction. In front of three fully connected layers is one global average pooling layer to impede overfitting. The hidden layers use a ReLU activation function.

The summary network condenses each LC with its complex physics information to a six-dimensional latent distribution. In the next section, we will see that the first stage of our training protocol pushes the network to identify this vector with the cosmological and astrophysical parameters

$$\theta = \{\Omega_{\mathrm{m}}, m_{\mathrm{WDM}}, T_{\mathrm{vir}}, \zeta, L_{\mathrm{X}}, E_0\} \ . \tag{8}$$

The cINN uses this physics-inspired latent LC representation as a condition for linking the six-dimensional model space to a Gaussian of the same dimensionality. Each of the six model parameters in Eq.(8) is normalized to the range $[0, 1]$.

The idea behind this setup is that the summary network extracts the relevant physics parameters from the complex data representation, a standard regression task, and that the cINN only needs to learn the correlations and the uncertainties for the posterior estimation. In case this intermediate representation of Eq.(8) is not optimal, the joint training of the summary network and the cINN corrects for a potential shortcoming to find the optimal latent representation. Depending on the downstream task, a higher dimensional latent space could offer additional flexibility. Here, we find the parameter-inspired choice to be sufficient for well-calibrated posteriors as shown in the following.

Nevertheless, this simple cINN task only requires a small number of simple coupling layers. We stack 8 affine coupling layers, where each of the internal fully connected networks

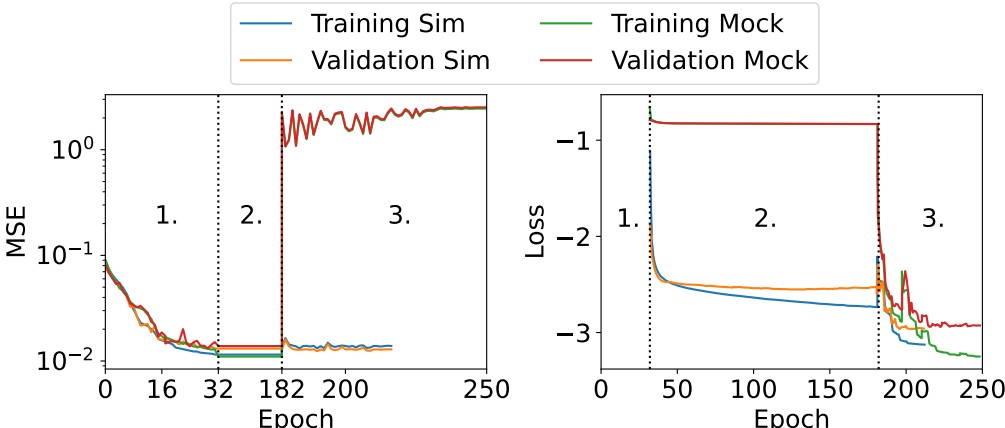

Figure 3: MSE of the summary network (left) and the cINN loss (right) during the three training stages, for training and validation datasets.

has 256 nodes with one hidden layer and ReLU activation. For the INN we use FrEIA (Framework for Easily Invertible Architectures) [66], all our training and evaluation is performed in PyTorch [67] with the Adam optimizer [68].

## 2.4 Training

For the training, we use 3600 simulated light cones, in 450 batches of eight. As the numbers of network parameters in the cINN and PIE-Net differ (#CNN parameters: 651526, #cINN parameters: 33488), the summary network needs to be pre-trained to provide a sensible basis for the cINN training. Without the pre-training step, the combination of both networks did not converge when jointly trained, probably due to the large difference in number of network parameters between the two networks. We use the physics information of the astrophysical and cosmological simulation parameters to set up a summary statistics in this region of the latent space. We train two different models, one on the pure noiseless simulations and one on the mock observations. We use a 3-stage training:

1. First, we pre-train the summary network as a regression network with an MSE-loss, since the parameter labels in Eq.(8) are known. The learning rate is initially set to $4 \cdot 10^{-4}$ and halved after 15 and 20 epochs. The training ends after 32 epochs. We do not require perfect convergence for this first stage.

2. Next, we only train the cINN with the pre-trained summary network. The learning rate is constant at $4 \cdot 10^{-4}$, and the training stops after 150 epochs, when the cINN starts to overfit. Adapting the learning right might slightly improve this pre-training, but without any effect on the final training.

3. Finally, both networks are trained together, so they can optimize the latent representation. The learning rate is reduced by a factor of 0.3 every five steps, starting from $4 \cdot 10^{-4}$, and stays constant after epoch 20. We stop the training after 32 epochs for the simulations and after 68 for the mocks.

Examples of the loss evolution for pure simulation and mock data in the three stages are shown in Fig. 3. The left panel shows the MSE between the true labels and the output of the summary network. During the first stage, it defines the training objective and therefore decreases to the point where training and validation loss slightly deviate. The second stage does not affect the summary network and its latent LC representation. In the third stage, when both networks are

trained with the cINN likelihood loss in Eq.(7), the MSE remains constant for the pure simulation and increases again for the mock data. The reason is that now the 6-dimensional output of the summary network can be corrected, away from the physical parameters given in Eq.(8), a more informative and unbiased summary can be found. For the pure simulation, such an adjustment is not needed. However, for mock data, the increased MSE indicates a significant change in the optimal latent representation when the two networks are trained jointly in stage 3. This difference is expected from a pure machine learning perspective — adding noise to the data makes it easier for the training to move around the loss landscape, including adjustments of the summary network required for optimal inference. We have checked that, indeed, the inference of cosmological parameters from mock data outperforms the pure simulations. This is why in the following we consider inference from pure simulations critical for understanding the behavior of the networks, but use the mock data to benchmark the 21cmPIE-INN performance.

The right panel in Fig. 3 shows the evolution of the cINN loss from Eq.(7). It is not defined in the first stage. In the second stage, it decreases and approaches the respective plateaus quickly. However, the loss values on the plateaus for the pure simulation and the mock data are different. Only when we also adjust the summary network in the third stage, the loss for the mock data reduces to the same level as for the pure simulation, indicating that at this stage both setups work at a similar level. As mentioned before, we still stick to the more realistic mock data performance whenever possible.

On an NVIDIA GeForce GTX 1080 Ti the training time per epoch is about 1 hour in stage 1, 1 minute in stage 2, and 1.3 hours in stage 3. These times reflect that in stages 1 and 3 the summary network is updated, so we need to read the LC dataset. As it is large (900 GB), we we cannot load it into memory and instead read the files for every batch. In stage 2 we use the fixed output of the summary network. The total training time is around 74 hours. The architecture and hyperparameters are summarized in Tab. 1. For the simulated mock light cones we train another set of identical networks in the same way.

## 2.5 Validation

To assess the network performance we use a set of standard metrics. First, we evaluate the parameter recovery. For a perfect posterior approximation, the mean of the marginalized posterior should statistically coincide with the true value of the parameter. To quantify the deviation, we use two metrics that measure if a sample of true parameters $\theta_j$ corresponds to a sample of estimated parameters $\hat{\theta}_j$. First, the coefficient of determination

$$R^2 = 1 - \sum_{j=1}^{J} \frac{(\theta_j - \hat{\theta}_j)^2}{(\theta_j - \bar{\theta}_j)^2} \tag{9}$$

measures the proportion of variance. The estimated parameters are the means of the sampled parameters and $\bar{\theta}$ denotes the mean of the true parameter samples. For perfect parameter recovery, we find $R^2 = 1$. Alternatively, the normalized root mean square deviation (NRMSE),

$$\text{NRMSE} = \frac{\sqrt{\frac{1}{J} \sum_{j=1}^{J} (\theta_j - \hat{\theta}_j)^2}}{\theta_{\max} - \theta_{\min}} \, , \tag{10}$$

includes a different normalization to make it scale-independent and to allow for comparison throughout all parameter ranges.

Second, simulation-based calibration (SBC) [69] is a self-consistency check to visually detect systematic biases. Given a sample from the prior $\tilde{\theta} \sim p(\theta)$ and one from the forward model $\tilde{x} \sim p(x|\theta)$ one can integrate out the $\tilde{\theta}$ and $\tilde{x}$ to recover the prior

$$p(\theta) = \int \mathrm{d}\tilde{x} \mathrm{d}\tilde{\theta} \, p(\theta|\tilde{x}) p(\tilde{x}|\tilde{\theta}). \tag{11}$$

This equation allows us to check for self-consistent sampling. If one samples from the correct posterior, Eq.(11) holds for any form of the posterior. Any violation of this equality indicates a problem in the sampling. If there is no violation then one can claim that the posterior is on average without bias and uses all the information from the summary. However, different biases in certain regions may cancel each other and the summary might not be optimal. The authors of Ref. [69] propose the SBC algorithm

---

1:    **for** $m = 1, \ldots, M$ **do**:
2:       Sample $\tilde{\theta}_m \sim p(\theta)$
3:       Simulate a dataset $\tilde{x}_m$
4:       Draw posterior samples $\hat{\theta}_l \sim p_\phi(\theta|\tilde{x}_m)$
5:       Compute rank statistic $r^{(m)} = \sum_{l=1}^{L} I_{[\hat{\theta}_l < \tilde{\theta}_l]}$
6:       Store $r^{(m)}$
7:    **end for**
8:    Create a histogram of $\{r^{(i)}\}_{m=1}^{M}$ and inspect for uniformity.

---

SBC requires a framework where fast amortized inference is possible, because many, typically up to a thousand or tens of thousands, posteriors need to be averaged. Therefore, this consistency check is only available in SBI and not for MCMC methods as it is computationally not feasible to perform this many MCMC runs for 21cm cosmology. Here, MCMC runs typically take weeks to months, as the signal is modelled based on simulations. SBC is already used in the field of Cosmology, see e.g. [70]. Intuitively, it counts how many sample values are larger than the true value. Normalized by the total number of samples, this gives the rank of the true parameter within this posterior. Over many iterations this should converge to a uniform distribution and visual discrepancies can be interpreted. If the histogram clusters at the extremes, ∪- or ∩-shaped, it shows over- or underconfidence, respectively. A slope in the histogram shows a bias.

Third, latent space examination allows us to trace the learned connection between model parameters and a unit Gaussian during the training. As training progresses, we can evaluate the cINN in the forward direction rather than the inference direction, to see if the latent distribution approaches a Gaussian shape.

Finally, the calibration error [71] quantifies how well the coverage of an approximate posterior matches the coverage of an unknown true posterior. For each parameter, the marginalized approximate posterior is given and the $\alpha$-credible intervals can be calculated, with $\alpha \in (0, 1)$. For each $\alpha$ the fraction of true parameter values lying in the interval is denoted with $\alpha_\theta$. A perfectly calibrated approximate posterior is given by $\alpha_\theta = \alpha$. The calibration error is then defined as

$$\mathrm{Err}_{\mathrm{cal}} = \sum_j |\alpha_{\theta,j} - \alpha_j|, \tag{12}$$

with equally spaced $\alpha_j$. A vanishing calibration error indicates perfect calibration.

## 3   Results

After training the 21cmPIE-INN in three stages, it can be used for inference, as illustrated in Fig. 1. This NPE setup extracts the full posterior by first reducing the dimensionality of

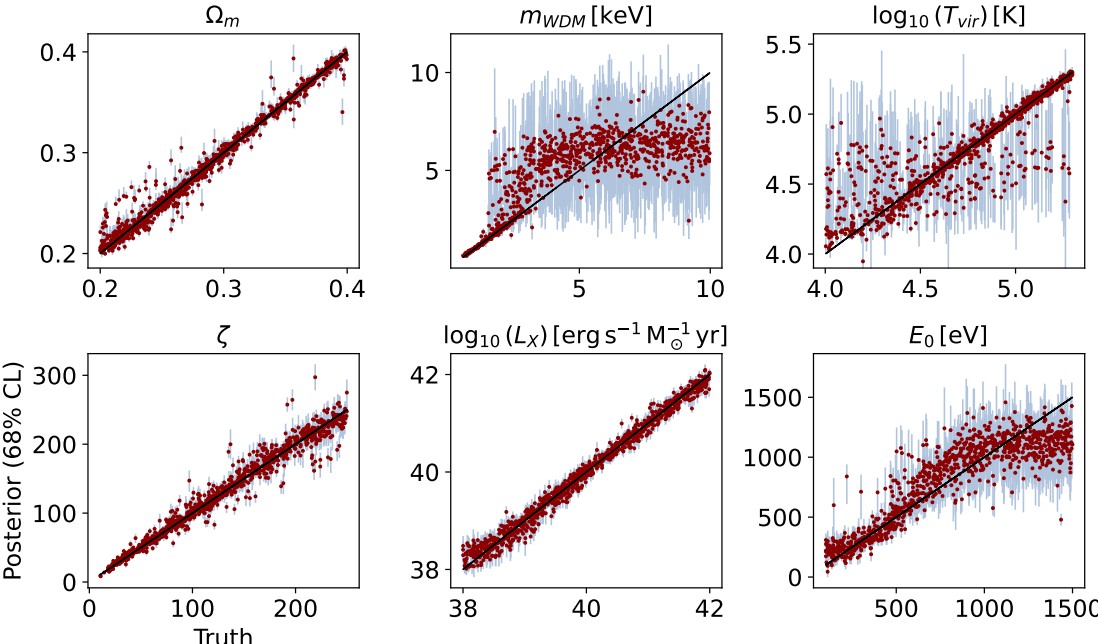

Figure 4: Calibration or parameter recovery for simulations, showing the mean of the marginalized posterior (red dots) and the 68% credible intervals (light blue).

the LC to a summary vector and then sampling from a Gaussian into model parameter space, conditioned on this summary vector. By examining the posterior and comparing it with the true labels, we can determine the performance of the network. We always discuss the results on the pure simulations first, followed by mock data including noise.

## 3.1 Performance and calibration

Before looking at the correlated posterior for the combination of cosmological and astrophysical parameters, we analyze the performance of the 21cmPIE-INN for the individual marginalized posteriors. In Fig. 4 we first show the recovered values for our six key parameters and their marginalized error bars from the cINN for pure simulation. The errors generally increase away from the diagonal line of perfect parameter recovery, as expected. However, for the different parameters we see a range of patterns, from near-perfect recovery to a significant fraction of outliers and the appearance of degeneracies.

First, the matter density $\Omega_{\mathrm{m}}$ is extracted almost perfectly, with small uncertainties and a diagonal calibration curve. In Tab. 2 we show the corresponding values for $R^2$, the NRMSE, and the calibration error of the approximate posterior.

The most difficult parameter to infer is $m_{\mathrm{WDM}}$, especially for large masses. First, for small $m_{\mathrm{WDM}}$ the network-derived parameter value is perfectly calibrated and very certain. For larger masses, the decreasing free-streaming length of WDM means that the signal looks more and more like CDM. This means we lose the relevant physics information, so the flat calibration curve is not a problem of the summary network or cINN. As a matter of fact, the huge error bars on the plateau account for this loss of information, and a likelihood analysis using power spectra comes to a similar result [72]. The metric to check for the coverage of the posterior, $\mathrm{Err}_{\mathrm{cal}}(m_{\mathrm{WDM}}) = 0.011$ is unexpectedly small, because it is not well-suited to capture this failure mode, where an overestimation for medium-sized $m_{\mathrm{WDM}}$ and an underestimation for large values cancel each other.

Table 2: Various performance metrics for the 21cmPIE-INN, shown for pure simulations and for mock data. $R^2$, NRMSE, and $\mathrm{Err_{cal}}$ are calculated according to Eq.(9), Eq.(10), and Eq.(12).

|  |  | $\Omega_{\mathrm{m}}$ | $m_{\mathrm{WDM}}$ | $T_{\mathrm{vir}}$ | $\zeta$ | $L_X$ | $E_0$ |
|---|---|---|---|---|---|---|---|
|  | $R^2$ | 0.981 | 0.621 | 0.764 | 0.969 | 0.987 | 0.803 |
| Simulation | NRMSE | 0.039 | 0.181 | 0.135 | 0.048 | 0.032 | 0.130 |
|  | $\mathrm{Err_{cal}}$ | 0.007 | 0.011 | 0.050 | 0.028 | 0.016 | 0.025 |
|  | $R^2$ | 0.990 | 0.663 | 0.716 | 0.969 | 0.973 | 0.045 |
| Mock | NRMSE | 0.028 | 0.171 | 0.149 | 0.048 | 0.047 | 0.286 |
|  | $\mathrm{Err_{cal}}$ | 0.045 | 0.017 | 0.086 | 0.066 | 0.025 | 0.064 |

For $T_{\mathrm{vir}}$ the truth is again recovered well, albeit with a small group of outliers. They are explained by a degeneracy with $m_{\mathrm{WDM}}$ [41]. A threshold for early star formation is set by $T_{\mathrm{vir}}$, but also by

$$M_{\mathrm{Jeans}} \propto (\Omega_{\mathrm{m}} h^2)^{1/2} \left( \frac{m_{\mathrm{WDM}}}{\mathrm{keV}} \right)^{-4} M_{\odot} \,. \tag{13}$$

For large $m_{\mathrm{WDM}}$ the Jeans mass limit becomes more important and the minimum virial temperature has little effect on the era of reionization, resulting in a degeneracy for $T_{\mathrm{vir}}$. However, the posterior for these parameter combinations is wide enough and does not underestimate the error budget.

Next, $\zeta$ and $L_{\mathrm{X}}$ show almost perfect parameter recovery with small network-derived error bars, as confirmed by high $R^2$ values, low NRMSE values, low calibration errors of the approximate posterior.

Finally, the recovery of $E_0$ degrades towards large values. The reason is that $E_0$ describes the threshold of self-absorption for host galaxies where this X-ray background is generated by compact X-ray binaries. Radiation below this threshold cannot escape the host galaxies. Our prior range is deliberately wide and motivated by the column density of the interstellar medium (ISM) in simulated high-redshift galaxies [73]. For large $E_0$, corresponding to high ISM column densities within high-redshift galaxies, only a small fraction of X-ray radiation can escape the galaxies, leading to a similarly small X-ray heating background for all scenarios above around 700 eV. The fact that high-$E_0$ scenarios become indistinguishable is reflected in the calibration curve and confirmed by the wide error bars.

One more performance test from Sec. 2.5, latent space examination is described in the Appendix. It checks consistency and convergence, giving more confidence in the network without having to know the true parameters. It confirms that the network is well-calibrated and that we are sampling from the correct posterior.

## 3.2 Mock data

It is important to check that our inference does not break down when the realistic data becomes noisy. This is why we repeat the performance and calibration study for the mock measurements introduced in Sec. 2.1. These more realistic results are shown in Tab. 2 and Fig. 5. The features, challenges, and patterns are similar to the pure simulation. Especially, the inference of the matter density $\Omega_{\mathrm{m}}$ remains very robust and almost perfectly calibrated. This aligns well with previous findings that even when transfer learning between different noise levels present in 21cm LCs a recovery of $\Omega_{\mathrm{m}}$ remains feasible [41].

The main physics effect of the noise added to the mock data is that the network now fails to infer $E_0$ altogether. This effect is completely unrelated to the effect of noise on the network training, discussed in Sec. 2.4. Because it removes information from the dataset, it

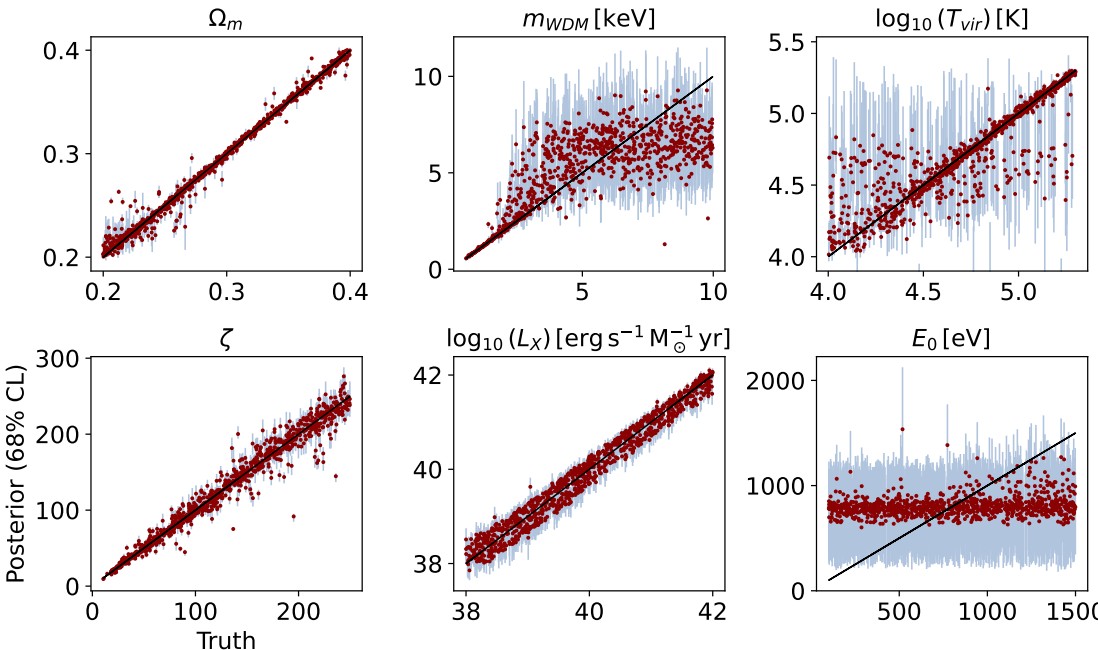

Figure 5: Calibration or parameter recovery for mock data, showing the mean of the marginalized posterior (red dots) and the 68% credible intervals (light blue).

has the opposite effect of the improved training performance, and both aspects have to be considered together. Confidence in the inference method is restored by the fact that the error bar on the posterior correctly accounts for this. If anything, the estimated error bars indicate that the network is slightly under-confident, as can be seen from the low calibration error $\text{Err}_{\text{cal}} = 0.025$.

The same bottom line can also be extracted from the simulation-based calibration in Fig. 6. We show histograms for a sample size of 10000 and 15 bins. Self-consistent sampling leads to a uniform distribution. The shaded region is the expected variation based on the 99% quantile of a Binomial. Common failure modes are overestimation, leading to a $\cup$-shape, and underestimation, leading to a $\cap$-shape. An asymmetry points towards a systematic bias. In our case, the histograms for the pure simulation show the expected variance, implying self-consistent sampling. After adding noise, all parameters except for $E_0$, and to a much smaller

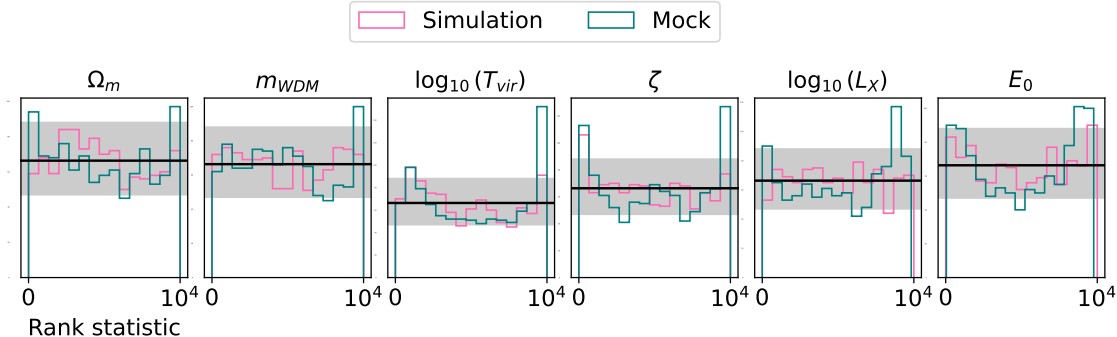

Figure 6: Simulation-based calibration, for pure simulation (pink) and mock data (teal). A uniform distribution indicates no bias, with the shaded region indicating the expected variation in uniform distributions, here the 99% quantile of a Binomial.

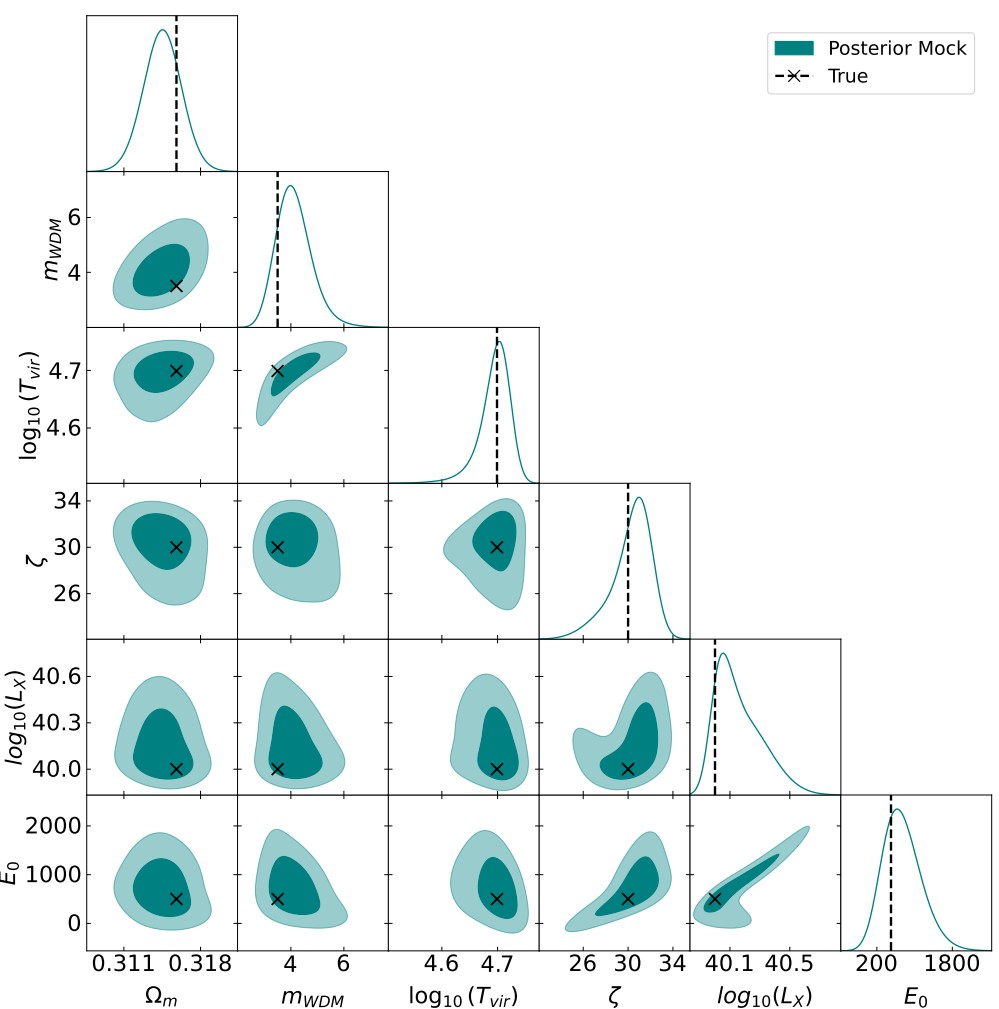

Figure 7: Marginalized posterior from mock measurements and for the light cone given in Eq.(14). The true values are shown as crosses. The shadings indicate 68% and 95% CI.

degree $\zeta$, remain well-calibrated. For $E_0$ the $\cup$-shape, implies under-confidence, corresponding to the issues observed in Fig. 5. Unlike standard MCMC analysis, where SBC is computationally not feasible, we can claim that our posteriors have the correct coverage and on average no bias.

## 3.3   Inference from 21cm light cones

Finally, we show the full posterior derived for one fiducial model from the corresponding simulated 21cm LC, with and without noise. The parameter values of our fiducial model are

$$\Omega_{\mathrm{m}} = 0.316\,, \qquad m_{\mathrm{WDM}} = 3.5\,\mathrm{keV}\,, \qquad T_{\mathrm{vir}} = 10^{4.70}\,\mathrm{K}\,,$$

$$\zeta = 30\,, \qquad L_X = 10^{40}\,\frac{\mathrm{erg/s}}{\mathrm{M}_\odot/\mathrm{yr}}\,, \qquad E_0 = 500\,\mathrm{eV}\,. \qquad (14)$$

Within the prior parameter ranges, any fiducial model parameter set can be chosen to extract the posterior fast from the trained model. Even though we sample the cINN 100.000 times, to ensure a reliable modeling of the tails of the multi-dimensional posterior, the analysis of a single fiducial takes only a few seconds. The numerical bottleneck is loading the data for the light cone.

The extracted posterior, with 2-dimensional correlations and 1-dimensional marginalized curves, is depicted in Fig. 7. We only show the realistic mock data, including noise. The fiducial parameter values fall within the $1\sigma$ region for all parameters. The posterior for the mock data comes out comparable to what we would get from the pure simulation. The parameter degeneracies expected from our discussion in Sec. 3.1 appear in the 2-dimensional correlations. For instance, $m_{\mathrm{WDM}}$ and $T_{\mathrm{vir}}$ show the degeneracies expected from Eq.(13). Also, the degraded inference of $E_0$ for the mock data can be traced to the strong degeneracy with typical X-ray luminosities $L_{\mathrm{X}}$. Most interestingly, the cosmological parameters and especially the matter density $\Omega_{\mathrm{m}}$ are inferred extremely robustly when it comes to adding noise. More results for different light cones are shown in App. B.

A direct comparison with other methods, such as a comprehensive and much slower MCMC analysis [74], is challenging, due to variations in the included noise models and set of parameters. However, our credibility intervals are qualitatively similar to those reported in Ref. [74].

# 4 Outlook

21cm experiments such as the SKA promise precise measurements of brightness temperature fluctuations of neutral hydrogen, offering a new tomographic perspective on the high-redshift universe and cosmological structure formation. To make optimal use of this complex and vast dataset, we need inference methods beyond simple summary statistics or power spectra. For this purpose, we developed a simulation-based inference method for a joint analysis of Cosmology ($\Omega_m$, $m_{\mathrm{WDM}}$), the Epoch of Reionization ($T_{\mathrm{vir}}$, $\zeta$), and Cosmic Dawn ($L_{\mathrm{X}}$, $E_0$). Our modern machine learning setup combines an advanced CNN summary network with a cINN to generate a multi-dimensional correlated posterior by sampling, linked by a physics-inspired latent representation of the complex SKA light cones.

We assessed the validity of this inference method through calibration curves and a range of metrics, coefficient of determination ($R^2$) and normalized root mean square error (NRMSE) for parameter recovery, as well as simulation-based calibration and calibration error. Aside from known correlations, which limit the possible inference, we found exceptional performance for pure simulation and for mock measurements with added noise. Only the energy threshold of self-absorption in galaxies $E_0$ becomes a challenge once noise is added to the dataset, becoming strongly degenerate with the typical X-ray luminosity as the second key CD parameter. Notably, the matter density $\Omega_{\mathrm{m}}$ as the key cosmological parameter is robustly inferred even in the presence of noise.

The summary vector that links the two networks of the 21cmPIE-INN is initialized to the parameters of interest, but adapted by the joint training with the cINN to guarantee an optimal inference, for the given underlying network. This optimization, the impact of the form and size of the summary vector, its stability, and its benefits in terms of explainable AI is an interesting avenue for further investigation. Similarly, our method allows for training on augmented datasets, for example with different experimental performance, opening further possibilities to analyze realistic SKA data.

Ultimately, the proposed approach facilitates rapid and straightforward simulation-based inference of likelihood constraints for cosmology. The generation of full posteriors from a given 21cm light cone takes a few seconds, most of this time used for reading in the complex data. This speed, combined with the ability to capture non-Gaussian information and learn optimal summaries (meaning well-calibrated with SBC and on average unbiased), distinguishes this method from alternative inference approaches.

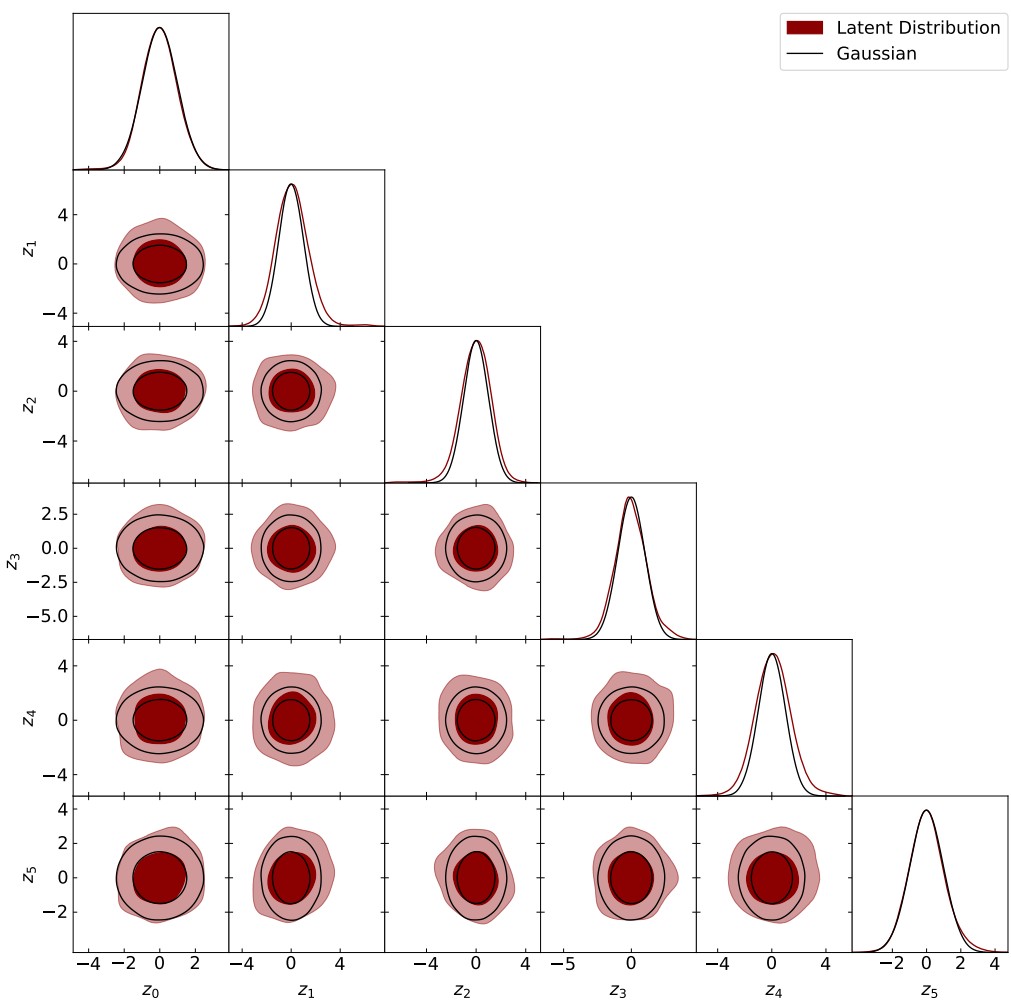

Figure 8: Distribution of the latent space variables and a unit Gaussian as comparison.

# Acknowledgments

We would like to thank Theo Heimel for extremely valuable help with INNs and Stefan Radev for his support in understanding all things BayesFlow.

**Funding information** BS is funded by Vector Stiftung. CH's work is funded by the Volkswagen Foundation. This work was supported by the DFG under Germany's Excellence Strategy EXC 2181/1 - 390900948 *The Heidelberg STRUCTURES Excellence Cluster.*

# A Latent space examination

As the learning objective of the cINN is to map the input to normal Gaussian distributions, for a converged network one would expect it to be distributed as such. This is quite easy to check visually, by sampling from the latent space and comparing it to a six-dimensional normal Gaussian. In our two cases of simulations-only and mocks with noise, both trained cINN networks pass this test. The distribution for the simulation-only case is shown in Fig. 8.

# B   Further posterior examples

The fast generation of the full posterior from one fiducial model allows a quick exploration of the parameter space within training prior ranges. To illustrate this fast inference from full 21cm LCs, we show three more posteriors for simulation and mock data in Figs. 9-11. The fiducial parameters are chosen at random and include combinations in the parameter space that are expected to be challenging for inference due to physical reasons, such as i.e. large $m_{\mathrm{WDM}}$. As can be seen from the marginalized 2D posterior contours the inference results are unbiased at the 1-2$\sigma$ level, both with and without noise, and independently of degeneracies or increased errorbars.

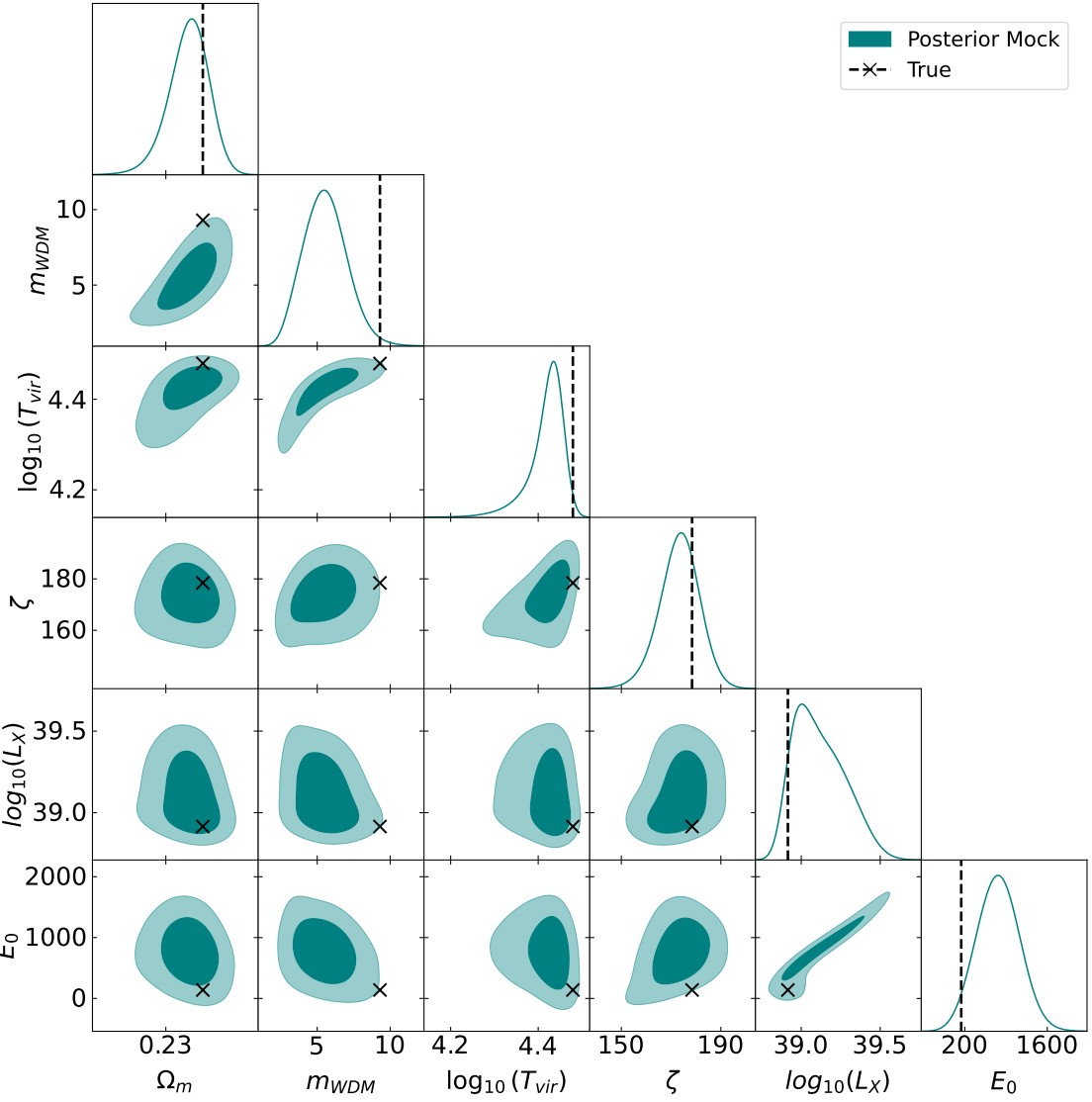

Figure 9: Marginalized 2D posterior and true values ($\Omega_{\mathrm{m}} = 0.237$, $m_{\mathrm{WDM}} = 9.30\,\mathrm{keV}$, $T_{\mathrm{vir}} = 10^{4.48}\,\mathrm{K}$, $\zeta = 178$, $L_X = 10^{38.9}\,\mathrm{erg\,s^{-1}\,M_\odot^{-1}\,yr}$, $E_0 = 139\,\mathrm{eV}$) known from the simulation. The contours are from a network trained and evaluated within an optimistic noise scenario for the SKA.

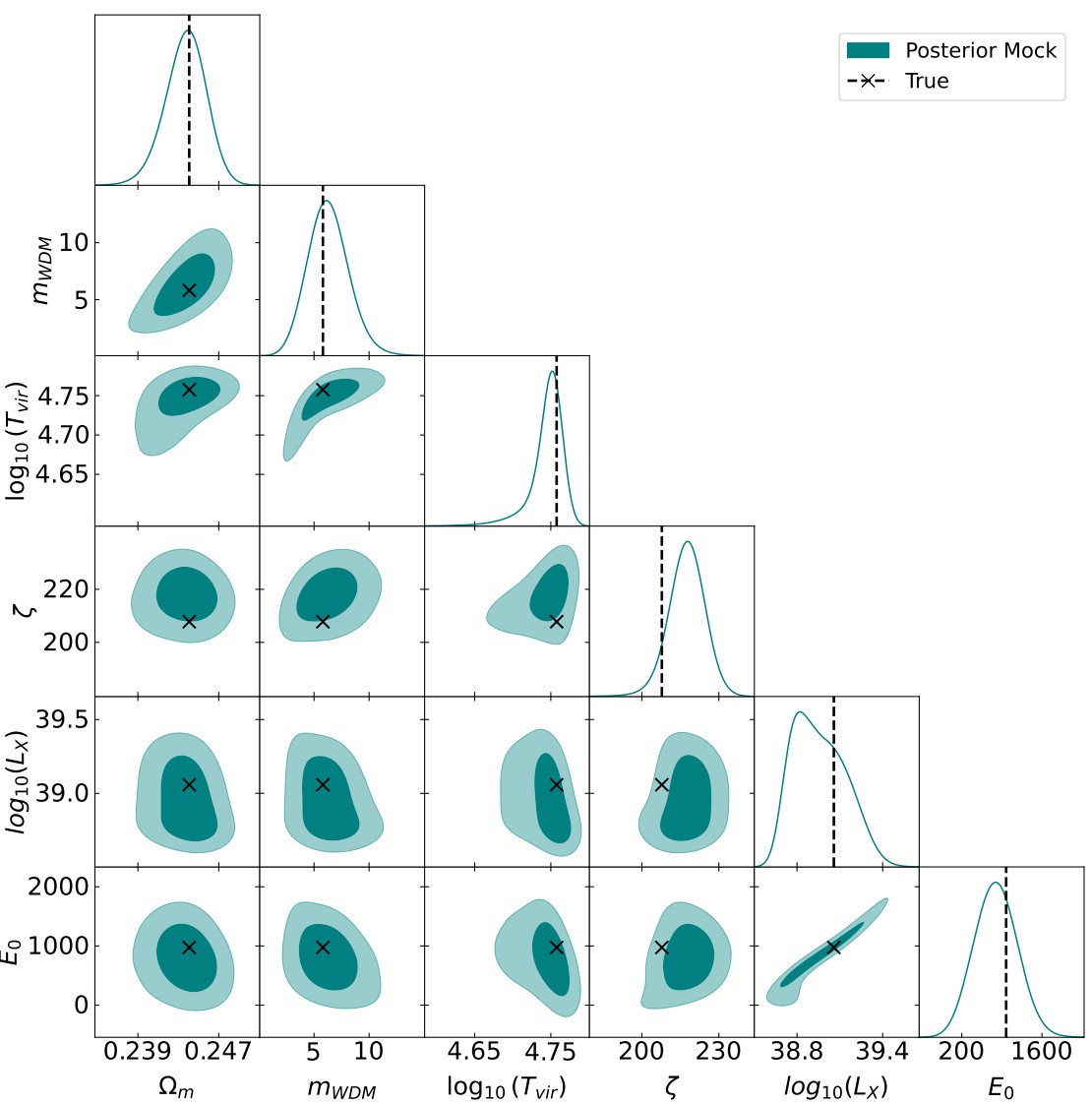

**Figure 10:** Marginalized 2D posterior and true values ($\Omega_{\rm m} = 0.244$, $m_{\rm WDM} = 5.81\,{\rm keV}$, $T_{\rm vir} = 10^{4.78}\,{\rm K}$, $\zeta = 208$, $L_X = 10^{39.1}\,{\rm erg\,s^{-1}\,M_\odot^{-1}\,yr}$, $E_0 = 974\,{\rm eV}$) known from the simulation. The contours are from a network trained and evaluated within an optimistic noise scenario for the SKA.

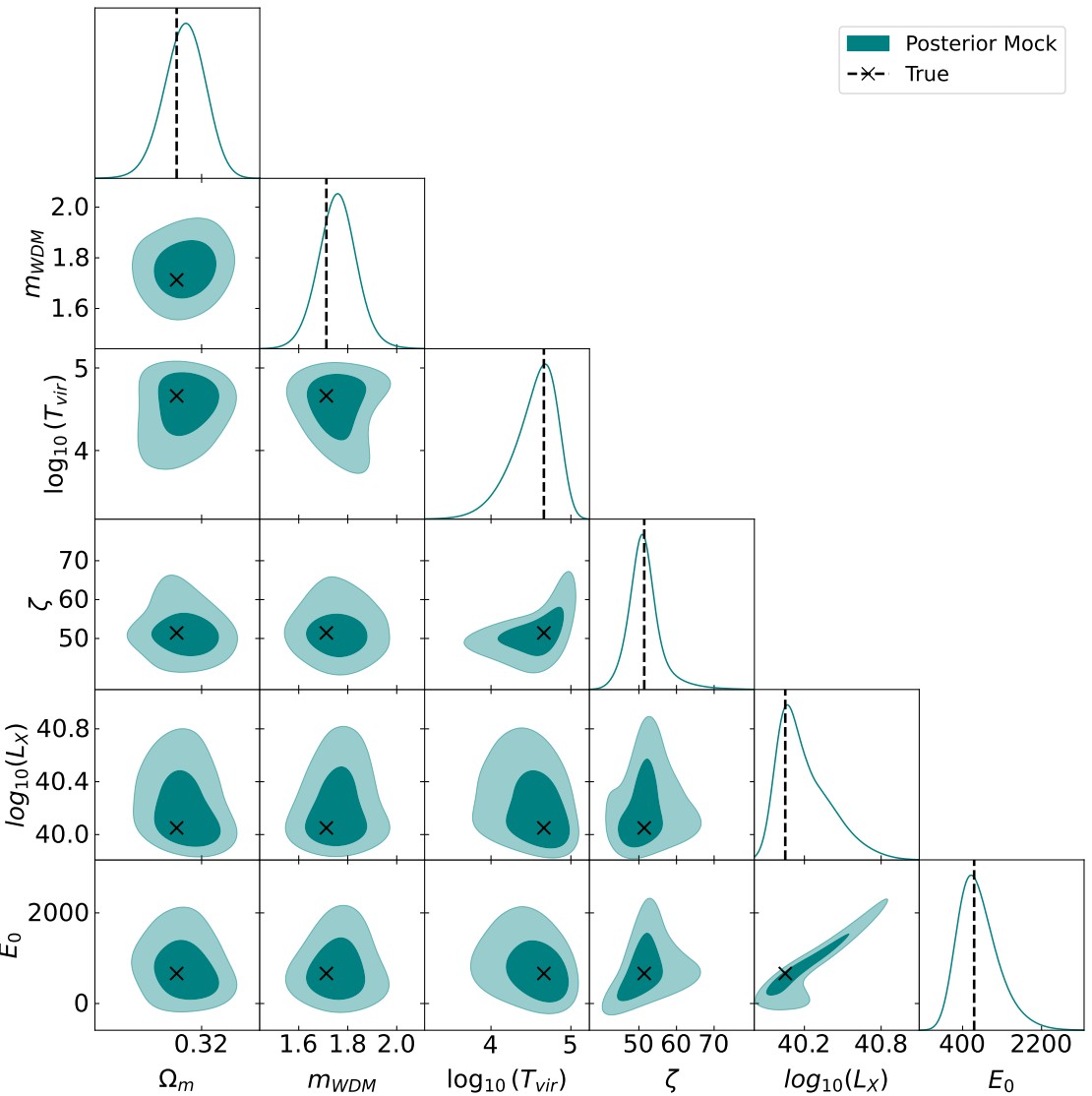

Figure 11: Marginalized 2D posterior and true values ($\Omega_{\mathrm{m}} = 0.318$, $m_{\mathrm{WDM}} = 1.71\,\mathrm{keV}$, $T_{\mathrm{vir}} = 10^{4.66}\,\mathrm{K}$, $\zeta = 51.4$, $L_X = 10^{40.0}\,\mathrm{erg\,s^{-1}\,M_\odot^{-1}\,yr}$, $E_0 = 663\,\mathrm{eV}$) known from the simulation. The contours are from a network trained and evaluated within an optimistic noise scenario for the SKA.

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
