# Peer review of "Optimal, fast, and robust inference of reionization-era cosmology with the 21cmPIE-INN"

_SciPost Physics Core, doi:SciPost Phys. Core 8, 037 (2025)_

## Round 2 · Referee Report · Anonymous (Referee 1) · 2025-3-19

Strengths

  1. Clear presentation of simulation codes and statistical approach
  2. Statistical soundness of proposed analysis
  3. Clear description of results

Weaknesses

  1. It remains unclear if the proposed approach gives better or faster results than a power-spectrum based analysis.

Report

I appreciate the authors’ efforts in addressing my previous comments and improving the paper’s presentation. Overall, I am satisfied with the revisions. However, I believe there is still some confusion regarding the meaning of optimality in this context and the role of SBC.

In their reply, the authors define optimality as obtaining posterior distributions that “use all of the information in the data and are well-calibrated.” They argue that SBC tests this and that passing the SBC test implies optimal information usage. However, this is not correct.

SBC only tests the consistency of the posterior given a particular data summary—here, the learned CNN summary. Even if the summary is entirely uninformative and the posterior is simply the prior, SBC would still be passed perfectly. This can be seen by replacing the posterior with the prior in the SBC algorithm cited in the paper. Thus, SBC cannot be used to claim that information is optimally used (i.e., that the CNN-derived summaries are sufficient).

Furthermore, this is distinct from the issue of underconfidence, which refers to posteriors being too wide given a specific data summary. SBC assesses whether posteriors are consistent with a given summary but does not evaluate the optimality of that summary itself.

Additionally, SBC tests posterior properties averaged over data realizations. It is possible that for some observations, posteriors exhibit positive bias, while for others, they are negatively biased—effects that may cancel out in the SBC histogram.

Otherwise, the paper is in good shape.

Requested changes

I would like to ask the authors to clarify the role of SBC in the context of optimality in both the abstract and the main text. They make strong claims about the optimality of their setup, but I do not see these fully supported by the tests presented in the paper. The paper is highly valuable and certainly deserves to be published as soon as possible. I also appreciate the use of SBC, which is an excellent method. However, SBC has specific caveats—it does not test the sufficiency of data summaries and may overlook varying biases—that should be explicitly described and properly reflected in the abstract, discussion and in the conclusions.

Recommendation

Ask for minor revision

  • validity: high
  • significance: high
  • originality: high
  • clarity: good
  • formatting: excellent
  • grammar: excellent

Author:  Benedikt Schosser  on 2025-03-21  [id 5303]

(in reply to Report 1 on 2025-03-19)

Dear Referee, thank you for the kind words, and we appreciate the feedback. We included the desired clarifications as well as reduced and clarified the usage of the word "optimal" in the paper. The exact modifications are in the list of changes.

---

## Round 2 · List of Changes

Introduction: Highlight more how our approach differs from previous ones (page 2)

Section 2: 2.1 - Fix typo (page 4)

2.2 -Rename section to Neural Posterior Estimation (page 4) -Remove most mentions of BayesFlow (whole paper)

2.3 -Discuss higher dimensional latent space (page 7&8)

2.4 - Give more information on training scheme (page 8) - More context on how the summary is adjusted (page 9)

2.5 - Discuss SBC much more (page 10)

Section 3:

3.2 Point out that SBC is not possible with MCMC (page 14)

---

## Round 3 · Referee Report · Anonymous (Referee 1) · 2025-3-24

Report

I'm happy with the changes made by the authors and recommend publication.

Recommendation

Publish (surpasses expectations and criteria for this Journal; among top 10%)

---

## Round 3 · List of Changes

• Change "optimal" to "efficient" in the abstract, clarifying the point of 'optimal' summaries.

  • In section 2.5 include the sentence: If there is no violation then one can claim that the posterior is on average without bias and uses all the information from the summary. However, different biases in certain regions may cancel each other and the summary might not be optimal.

  • In section 3.2 include "on average" in the following discussion: Unlike standard MCMC analysis, where SBC is computationally not feasible, we can claim that our posteriors have the correct coverage and on average no bias.

  • In the outlook, we clarify that optimality is reached by joint training, given the network architecture we choose: The summary vector that links the two networks of the 21cmPIE-INN is initialized to the parameters of interest, but adapted by the joint training with the cINN to guarantee an optimal inference, for the given underlying network.

  • In the outlook we clarify what we mean by optimal in the following sentence: This speed, combined with the ability to capture non-Gaussian information and learn optimal summaries (meaning well-calibrated with SBC and on average unbiased), distinguishes this method from alternative inference approaches.

---

## Editorial Decision

published